# Realistic turbulent inflow conditions for estimating the performances of a floating wind turbine

Cédric Raibaudo[1,2], Jean-Christophe Gilloteaux[1], and Laurent Perret[1]

[1]Nantes Université, Centrale Nantes, CNRS, LHEEA, UMR 6598, F-44321 Nantes, France
[2]University of Orléans, INSA-CVL, PRISME, EA 4229, 45072, Orléans, France

**Correspondence:** Laurent Perret (laurent.perret@ec-nantes.fr)

**Abstract.** A novel method for generating turbulent inflow boundary conditions for aeroelastic computations is proposed, based on interfacing hybrid hot-wire and particle image velocimetry measurements performed in a wind tunnel to a full-scale load simulation conducted with FAST. This approach is based on the use of the proper orthogonal decomposition (POD) to interpolate and extrapolate the experimental data onto the numerical grid. The temporal dynamics of the temporal POD coefficients is driven by the high-frequency hot-wire measurements used as inputs of a lower-order model built using a multi time-delay linear stochastic estimation (LSE) approach. Being directly extracted from the data, the generated 3-component velocity fields later used as inlet conditions present correct one- and two-point spatial statistics and realistic temporal dynamics. Wind tunnel measurements are performed at a scale of 1:750, using a properly scaled porous disc as a floating wind turbine model. The wave-induced motion of the platform is imposed with a linear actuator. Between all 6 degrees-of-freedom (DoF) possible, the present study focus on the streamwise-direction motion of the model (surge motion). The POD analysis of the flow, with or without taking into account the presence of the surge motion of the model, shows that a few modes are able to capture the characteristics of the most energetic flow structures and the main features of the wind turbine wake such as its meandering and the influence of the surge motion. The interfacing method is first tested to estimate the performance of a wind turbine in an offshore boundary layer and then those of a wind turbine immersed in the wake of an upstream wind turbine subjected to a sinusoidal surge motion. Results are also compared to those obtained using the standard inflow generation method provided by TurbSim available in FAST.

## 1 Introduction

In recent years, floating offshore wind turbines (FOWT) have been of particular importance for both scientific and industrial applications. Mounted on a floating structure, these turbines can be installed in deep waters, with wider space and more available power, compared to onshore and fixed offshore wind turbines (Cruz and Atcheson, 2016). Understanding wind turbines' wakes is crucial to improve their aerodynamics when operating in farms (Schmidt and Stoevesandt, 2015; Bastine et al., 2018). For FOWT embedded in farms, multi-dynamics and multi-scale phenomena can impact the wake characteristics and therefore the farm performances (Huang et al., 2018). In particular, the wake interaction between FOWTs significantly affects the power produced by the farms and needs to be considered carefully (Huang et al., 2018).

For the modelling and the prediction of the wind turbine wake, simplified analytic models have been suggested based mostly on the mean velocity deficit (Jensen, 1983; Frandsen et al., 2006; Bastankhah and Porté-Agel, 2014). These models could lack of precision but are easier to compute, compared to high-cost simulations, especially for wind farms control. Other parameters such as the turbulence intensity, the geometrical properties (turbine diameter $D$, position in the wake $x/D$, roughness length $z_0$, etc.), the thrust, can be included in the models for better representativity (review of Kaldellis et al. (2021) for more details).

Porous discs have been used as surrogates for wind turbines for wind turbines studies. This approach is similar to the actuator disc theory used for theoretical and numerical prediction of wind turbine performances, in which the rotor is replaced by a permeable disc generating an equivalent thrust (Betz, 1920; Joukowsky, 1920). Pioneer study of Aubrun et al. (2013) showed that the mean velocity deficit, the integral scales and the statistics of the porous disc wake are similar to those of a model with rotor for downstream positions in the wake $x/D > 3$. Similar conclusions were found by Camp and Cal (2016) for $x/D > 3.5$. Porous discs are therefore valid for the characterization of the far-field wake, where, in farms, other downstream wind turbines are expected to be located (Porté-Agel et al., 2020).

Unsteady phenomena occur in the wind turbine wake, and in particular the wake meandering. Meandering is a large-scale motion of the wake due to the interaction between the rotor and large coherent structures of the incoming boundary layer (Porté-Agel et al., 2020; Hodgkin et al., 2023). The meandering is also sensitive to the incoming flow conditions, such as the turbulence level (Foti et al., 2018; Porté-Agel et al., 2020). The physical mechanism of this phenomenon is still not fully understood today, and continue to be an object of study.

For FOWTs, the unsteady behavior of the wake interacts with the wave-induced motion of the platform. Rockel et al. (2014) firstly studied the effect of one of the motion component - the pitch - on the wake dynamics with stereoscopic Particle Image Velocimetry (PIV). The study showed that the motion creates additional vertical velocity and reduces the kinetic energy available for a downwind turbine. Khosravi et al. (2015) considered the surge motion and also observed a slow-down of the wake recovery and a decrease of the kinetic energy, but an increase of 1 % of power produced for the given turbine. By considering pitch and roll experimentally, Fu et al. (2019) showed no effect of the motions on the mean power produced. However, the pitch motion reduces the power fluctuations in the lowest frequency part of the spectrum, for frequencies lower than the imposed pitch motion harmonics. The roll motion also reduces these fluctuations around a specified frequency, about the half of the motion main harmonics. On the other hand, Corniglion et al. (2020) observed the effect of a high-frequency surge motion, in particular an increase of the mean thrust and the axial velocity in the wake. Recent works (Messmer et al., 2022; Fontanella et al., 2022; Chen et al., 2022; Li et al., 2022) investigated the effect of more complex motions on the wake dynamics and the meandering in particular. However these effects are still not fully understood for floating wind turbines. These difficulties of understanding increase when considering realistic atmospheric conditions for numerical and experimental studies. By varying the turbulence intensity of the incoming flow, significant differences in the wake properties and/or the power produced by the wind turbine can be observed (Saranyasoontorn and Manuel, 2005; Chamorro and Porté-Agel, 2009; Nybø et al., 2020). Other studies (Chamorro and Porté-Agel, 2009; Bastine et al., 2014; Schliffke, 2022) considered significant atmospheric turbulent boundary layer upstream the wind turbine. Axisymmetry of the incoming flow in particular plays an important role on the redistribution of the turbulence levels in the turbine wake (Chamorro and Porté-Agel, 2009).

As explained in the two previous paragraphs, taking into account realistic conditions of the incoming flow and wave-induced motions is crucial to estimate the performances of floating wind turbines. It is also clear today that atmospheric turbulence cannot be treated as a purely random phenomenon characterized only by its intensity. To address this problem, the strategy commonly used in the wind energy community is based on the generation of synthetic turbulence with a target spectral content representative of the atmosphere (Kelley and Jonkman, 2007; Mann, 1998). This approach, implemented in the TurbSim turbulence generator of the FAST aeroelastic code of the National Renewable Energy Laboratory (NREL), has been improved to take into account particular phenomena such as low-level jets or Kelvin-Helmholtz instabilities that can develop in the atmosphere under certain conditions (Kelley and Jonkman, 2007). In the case of wind farm studies, the turbines are located in the wake of other machines; the synthetic turbulence approach can no longer be used directly and must be modified to take into account the presence of the wake so as to perform aeroelastic calculations under realistic realistic conditions. The developed approaches range from simple wake models imposing an additional velocity deficit to stochastic wake models based on LES (Large Eddy Simulation) type precursor computations through intermediate models taking into account the velocity deficit but also the meandering of the wake (see Bastine et al. (2018), and references herein).

In the context of LES computation of turbulent flows, Perret et al. (2006, 2008) proposed an alternative method to generate realistic unsteady inflow conditions by coupling wind tunnel measurements to an LES code to simulate a turbulent mixing layer flow. In their approach, non-time resolved stereoscopic PIV measurements were performed in a cross-section of the flow corresponding to the inlet section of the targeted LES simulation. Using Proper Orthogonal Decomposition (POD) to decompose the velocity fields into a set of spatial modes and temporal coefficients, a low-order representation of the flow was built. The time evolution of the temporal coefficients was modeled either in a purely stochastic manner via random Gaussian numbers with a realistic spectrum (Perret et al., 2006) or by deriving a low-order dynamical model of the temporal evolution of the most energetic structures (Perret et al., 2008). The incoherent motion was modeled by employing time series of Gaussian random numbers to mimic the temporal evolution of higher order POD modes. Beyond the fact that this innovative method allows for the study of complex flow configurations (as long as wind tunnel measurement can be performed), it enables the generation of inlet conditions with a realistic spatio-temporal organization, a feature often missing in purely stochastic approaches.

Building upon the work of Perret et al. (2006, 2008), the present study aims to propose a novel approach based on a direct coupling between experimental data from a scaled-down wind tunnel experiment and a full-scale load computational code (FAST from NREL in the present case). The methodology is applied on experimental data representative of a FOWT model under wave-induced surge motion and realistic conditions of the incoming flow. As detailed in the following, it relies on stereoscopic PIV performed in a cross-section of the wind-turbine wake combined with the derivation of a data-driven low order model to improve the temporal dynamics of the PIV measurements. This database of well time-resolved vector fields generated in the cross-section of the flow are then used as unsteady inlet conditions for the load computation. After a description of the experimental setup (§2), details on the scaling between the model and a real scale prototype will be discussed (§3). The flow without model and the mean wake are then presented (§4). The velocity fields are reconstructed at high-sampling frequency using stochastic estimation (§5), before being coupled to FAST simulations to evaluate the power produced by a downwind floating wind turbine submitted to the wake under surge motion (§6).

 **2  Experimental setup**

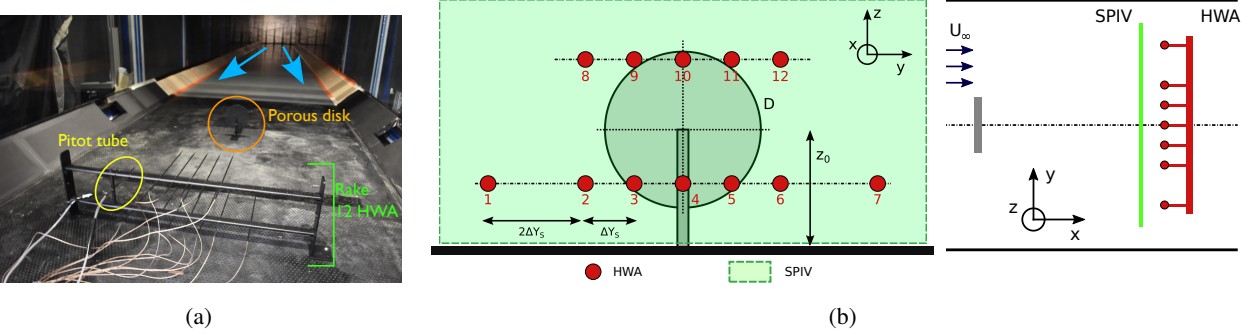

(a)                                                          (b)

**Figure 1.** Experimental setup: (a) picture of the setup in the wind-tunnel and (b) schemes of the metrology setup.

Experiments were carried in the LHEEA lab's atmospheric wind tunnel. Without model, the free-stream velocity of the wind is $U_\infty = 4.2$ m/s and its turbulent intensity far from the wall about 0.5%. The experimental setup is presented in Fig. 1. The wind turbine is modeled using a porous disc of diameter $D = 0.16$ m. The disc center is set at a height $z_{hub} = 0.12$ m with respect to the floor. The model represents the floating 2MW wind turbine used in the FLOATGEN research project, installed at Centrale Nantes' offshore test site in Le Croisic, France (Rousset et al., 2010). The thrust coefficient $C_t$ is estimated to be approximately 0.65 and the power coefficient $C_p \approx 0.25$ (Schliffke et al., 2020). For streamwise positions $x/D$ in the wake further than 3, the use of porous discs for the characterization of wind turbines wakes is considered valid (Schliffke et al., 2020). The thickness of the incoming boundary layer is $\delta = 0.6$ m and the resulting Reynolds number is $Re = U_\infty \delta / \nu \approx 160\,000$.

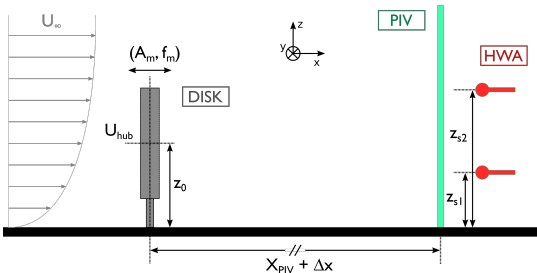

**Figure 2.** Sketch of the wave-induced surge motion experimental setup.

To replicate realistic behaviors of floating wind turbines under wave swell, a sinusoidal surge-motion is imposed on the model using a linear actuator (Schliffke et al., 2020). The setup to model this motion is presented in Fig. 2. The sinusoidal amplitude is $\pm$ 0.01 m and the frequencies tested $f_m = [0, 2, 3, 3.75]$ Hz. Using the incoming velocity speed and the disc diameter, it corresponds to a non-dimensionalized frequency $F_m^+ = f_m D / U_\infty = [0, 0.08, 0.11, 0.14]$. The model position is controlled retroactively and monitored, with a bias of about 1 mm between the expected and real positions (Schliffke et al.,

2020). As indicated in Fig. 2, the variable x-axis corresponds to the flow direction, the $y - z$ plane is normal to the flow. The coordinates origin is chosen at the disc center.

Instantaneous three-components velocities of the wake were acquired using stereoscopic PIV in a $y - z$ plane normal to the main flow direction, at two streamwise positions $x/D = 4.6$ and $8.1$ downstream of the wind turbine model. Two $2560 \times 2160 px^2$ sCMOS Zyla cameras from Oxford Instruments were set upstream on each side of the wind tunnel. Nikon lens with a focal of $f_o = 60$ mm were fixed on the cameras. Scheimpflug adapters were used to ensure a focused image over the full field of view. The laser sheet was generated using a double pulse Nd-Yag laser (200 mJ per cavity). A LaVision Laskin-Nozzle aerosol generator was used to seed with olive oil droplets of 1 $\mu$m diameter. Transparent film was added at the walls to reduce the light diffusion. The time delay between the two pulses was $dt_{PIV} = 300$ $\mu s$. Image analysis was performed using a standard cross-correlation multi-pass algorithm with a final interrogation window size of $32 \times 32$ $px^2$ using Dantec Dynamics software. The size of the final region of interest is about $3D \times 2D$. The sampling frequency is $f_{PIV} = 14.1$ Hz and $14000$ snapshots were acquired for each case. The number of snapshots was doubled for one case to estimate the statistical convergence.

Constant temperature anemometry (CTA) measurements were also performed simultaneously to the PIV. Twelve hot-wire (HWA) sensors were distributed spanwise, downstream the PIV measurement plane and at two heights ($0.47D$ and $1.25D$). The sampling frequency for the sensors was $f_{HWA} = 15$ kHz. Sensors were calibrated using King's law by measuring the free-stream flow with the help of a Pitot tube. The calibration procedure and the post-processing of the raw voltages accounted for temperature correction using the method proposed by Hultmark and Smits (2010).

## 3 Scaling

The experimental setup presented in the previous section has been chosen to respect scaling properties between the model and the chosen prototype. A complete study on how to scale wind turbines rotors has been developed by Canet et al. (2021) for example, and a full explanation of the scaling methodology for the present case can be found in Schliffke (2022). As the numerical model of the FLOATGEN prototype or its detailed characteristics are not available for the FAST simulations due to confidentiality constraints, the 5MW reference wind turbine is used (Jonkman et al., 2009). The full-scale rotor diameter is $D_P = 120$ m, the hub height $z_{0,P} = 90$ m, the atmospheric boundary layer about $\delta \approx 300$ m and the incoming velocity at the hub height $U_{0,P} = 10$ m/s, also corresponding to the parameter implemented in the present FAST simulations.

Therefore, scaling factors are determined between the model and the NREL reference wind turbine. First, the geometric scaling factor based on the rotor diameter is $\Lambda_L = D_P/D = 750$. The porous disc diameter is chosen to respect a blocking factor of 0.5 %, which is under the 5 % required by the VDI guideline 3783 (VDI, 2017). The model hub height is fixed at $z_0 = 0.12$ m to be totally immersed in the boundary layer of height $\delta = 0.6$ m. It corresponds to the NREL hub height of 90 m immersed in atmospheric boundary layers of values between 300 and 500 m. The integral scale of the flow is about $0.4$ m, corresponding to an integral scale of 200 m for the full scale, also respecting the VDI guidelines (VDI, 2017; Schliffke, 2022).

The velocity scaling factor is defined using the hub velocity by $\Lambda_V = U_{hub,P}/U_{hub}$. The standard objective for FAST simulations is $U_{hub,P} = 10$ m/s. For the model, for a freestream velocity of $U_\infty = 4.2$ m/s, the velocity at hub height in the

undisturbed oncoming boundary layer is $U_{hub} = 3.8$ m/s. The scaling factor is therefore $\Lambda_V = 2.6$. Using Strouhal number similarity, the time scaling factor is $\Lambda_t = \frac{\Lambda_L}{\Lambda_V} = 284$. This means physical phenomena in the wind tunnel occurs 284 faster than for the prototype. For the FAST implementation with this time factor, to simulate the aerodynamics at a classical sampling

frequency around 20 Hz, velocity fields are therefore needed at a minimal frequency of 5680 Hz. Reconstruction of the velocity fields at a frequency higher than that of the PIV measurements is therefore necessary.

The displacement system has been designed to replicate measurements performed on the SEM-REV test site where the FLOATGEN prototype. The design justification is fully explained in Schliffke (2022), based on the study of Tarpin (2018). The most probable sea state is of height $H_S = 4.6$ m and characteristic wave period $T_P = 11$ s. Considering the wind turbine

properties, it leads to a most probable wave motion of $F^+ = 0.7$. But, as explained by Schliffke (2022), the operating frequency range is between $F^+ = 0.4 - 3.2$. The chosen actuator frequencies are therefore low compared to the expected wave motion of the full scale wind turbine. However, it covers part of the low-frequency range of the natural frequencies and of the same order of magnitude as the expected phenomena.

## 4 Mean structure of the flow

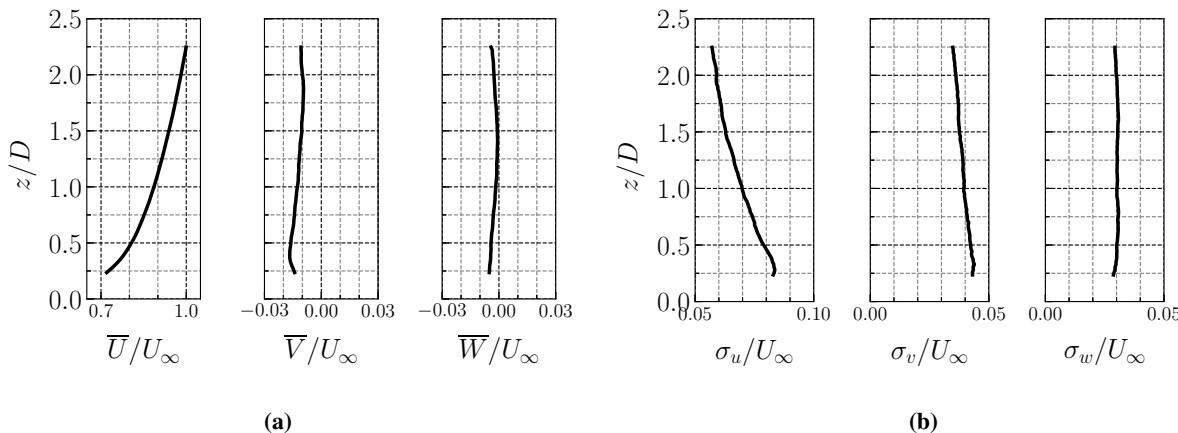

|       |       |
|:-----:|:-----:|
| **(a)** | **(b)** |

**Figure 3.** Mean $\overline{U}/U_\infty$ (a) and RMS velocities $\sigma_u/U_\infty$ (b) at $y/D = 0$ for the three velocities components for the turbulent boundary layer without model. Different scales chosen between the streamwise velocity and the other velocity components.

The turbulent boundary layer without model is first documented with the present experimental campaign. Only the main elements of the flow without model are presented here, more details can be found in previous works (Perret et al., 2006, 2008; Schliffke et al., 2020; Raibaudo and Perret, 2022). Mean and RMS velocities of the atmospheric turbulent boundary layer without model are presented in Fig. 3. Profiles of the three components along the wall-normal direction $z/D$ going through the disc center ($y/D = 0$) are considered here. Mean streamwise velocity $\overline{U}/U_\infty$ corresponds to a classical turbulent boundary

layer profile. Near the wall, the RMS streamwise velocity reaches $\sigma_u/U_\infty = 0.082$, leading to a maximum turbulence level

$I_u = \sigma_u/\overline{U} \approx 13$ %. A small negative spanwise velocity $\overline{V}/U_\infty \approx 0.01$ is also observed in the figure. It has been observed in previous works and could be linked to large flow structures developing upstream the measurement region.

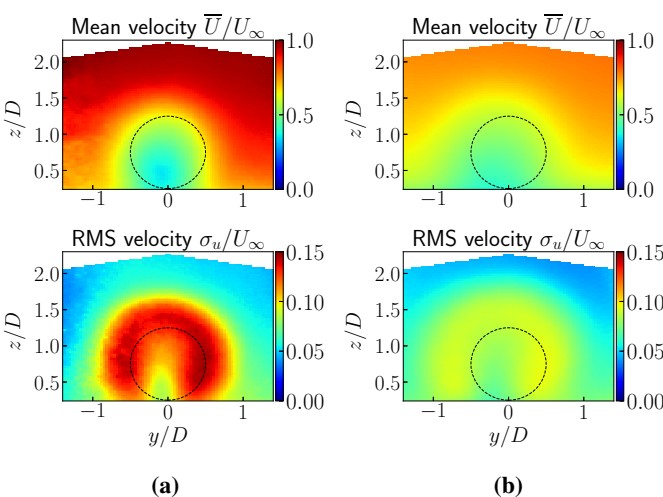

**Figure 4.** Streamwise mean $\overline{U}/U_\infty$ (first row) and RMS velocity $\sigma_u/U_\infty$ (second row) for: (a) wake for the fixed model at $x/D = 4.6$ and (b) wake for the fixed model at $x/D = 8.1$. Dashed lines correspond to the position of the porous disc in the $Y - Z$ plane.

The mean wake flow is then presented here. Streamwise mean and RMS velocity fields are presented in Fig. 4. Two streamwise positions in the wake are presented here: $x/D = 4.6$ and 8.1. The mean flow in the near-wake is disturbed by the disc, following its shape (represented with dash line in Fig. 4). At $x/D = 8.1$, the wake center is slightly shifted to the negative $y$-direction, also seen in previous works (Schliffke et al., 2020; Foti et al., 2019). It could correspond to the small spanwise velocity of the incoming flow observed previously (Raibaudo and Perret, 2022). Compared to the flow without model, the turbulence intensity increases in the near-wake from 8 % up to 15 %. The circular horseshoe shape in the near-wake has been found in previous works (Hamilton et al., 2018). The slight shift in the negative $y$-direction is observed in the RMS fields at $x/D = 8.1$. It must be noted here that, at the most downstream location $x/D = 8.1$, two distinct peaks in the horizontal profile of the RMS of the streamwise velocity component measured at hub height are still visible, indicating that the wake is not fully developed yet. However, the recovery of the mean velocity $u(z = z_{hub})/U_{hub}$ reaches 60 % of the undisturbed velocity, with a corresponding decreasing turbulent intensity. The wake is therefore still in its transition phase towards a fully developed wake. More details on the mean flow can be found in Schliffke et al. (2020) and Belvasi et al. (2022).

## 5   Stochastic reconstruction using POD-LSE multi-delays

For the coupling of wind-tunnel PIV velocity fields with an aeroelastic code to perform a full-scale load simulation, due to the scaling constraints presented in §3, high-sampling frequency is required. By combining the velocity fields acquired with

PIV and time-resolved HWA measurements, stochastic approaches are used here to reconstruct velocity fields at high temporal resolution.

## 5.1 Methodology

Reconstruction of the velocity fields is performed using a Proper Orthogonal Decomposition and multi-time-delay Linear Stochastic Estimation (POD-mLSE) approach (Durgesh and Naughton, 2010), which is an extension of the standard POD-LSE technique developed by Bonnet et al. (1994). Velocity fields are first decomposed into temporal and spatial modes using the standard snapshot POD (Sirovich, 1987):

$$\mathbf{u}(\mathbf{x},t) \approx \sum_{i=0}^{N_m} a_i(t)\, \Phi_i(\mathbf{x}) \tag{1}$$

with $N_m$ the number of modes selected for the truncature. The POD procedure is not fully detailed here, but the main elements are precised now. A kernel $\kappa$ is build from the snapshots matrix $\mathbf{U_s}$ corresponding to the velocity fields reorganized as 1D-vectors for each time: $\kappa = \mathbf{U_s^T U_s}$. The square matrix $\kappa$ is therefore inverted to solve an eigenvalue problem:

$$\kappa\, A_i \;=\; \lambda_i\, A_i \tag{2}$$

with $\lambda_i$ the eigenvalues and $A_i$ the temporal modes matrix in which the temporal modes $a_i$ are extracted. After a descending sort by eigenvalues amplitude, the spatial modes $\Phi_i$ are obtained as:

$$\Phi_i \;=\; A_i U_s \,/\, ||A_i U_s|| \tag{3}$$

Here, $N_m = 100$ modes are chosen, corresponding to about 80% of the cumulative eigenvalues energy. The objective of the stochastic estimation is to determine the more suitable coefficients $B_{ijk}$ to express the relation between the temporal modes $a_i$ and the HWA sensors with time delays $\tau_k$: The reconstructed temporal modes $\widehat{a}_i$ are then estimated at higher sampling rate:

$$\widehat{a}_i(t) \;=\; \sum_{j=0}^{N_j} \sum_{k=0}^{N_k} B_{ijk}\, S_j(t - \tau_k) \tag{4}$$

with $N_j = 12$ the number of HWA sensors $S_j$ and $N_k$ the number of delays imposed on the sensors. Here, $N_k = 21$ delays are distributed between $\pm 0.2$ s, following comparable reconstruction parameters on similar experiments (Blackman and Perret, 2016). The coefficients $B_{ijk}$ are determined through the resolution of a least-square minimization problem using cross-correlations:

$$B_{ijk} \;=\; \mathcal{C}_{\{a_i,\, S_{jk}\}}^{-1}\, \mathcal{C}_{\{S_{jk}\}} \tag{5}$$

with $\mathcal{C}_{\{a_i,\, S_{jk}\}}$ the cross-correlations between the temporal modes and the sensors with delays, and $\mathcal{C}_{\{S_{jk}\}}$ the correlations between the different sensors with different delays. Temporal modes $\widehat{a}_i$ are therefore estimated at higher temporal resolution using Eq. 4, then combined with the spatial modes using POD (Eq. 1) to reconstruct the velocities. Only results for the downstream location $x/D = 8.1$ are presented here, which corresponds to the inlet section of the load simulations. The sampling frequency used for the reconstruction is $f_s = 7.05$ kHz, to meet the conditions required for these simulations.

## 5.2 Proper Orthogonal Decomposition

A selected set of results on the POD analysis is presented here, but more details can be found in previous works from the authors (Raibaudo et al., 2022).

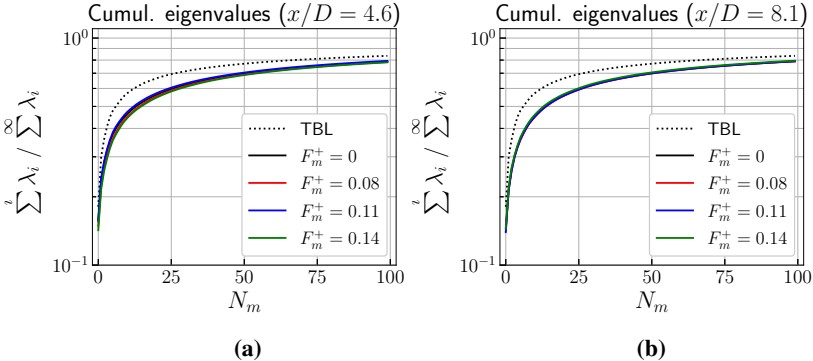

**Figure 5.** Cumulative eigenvalues $\sum\limits^{N_m}\lambda_i/\sum\limits^{\infty}\lambda_i$ from the POD obtained from different streamwise positions (a) $x/D = 4.6$ and (b) $x/D = 8.1$. For each configuration, four frequencies of surge motion $F_m^+$ are presented in solid lines: 0 (fixed), 0.08, 0.11 and 0.14. TBL without model is presented in dotted line.

The convergence using the cumulative eigenvalues $\sum\limits^{N_m}\lambda_i/\sum\limits^{\infty}\lambda_i$ is first shown in Fig. 5. The two streamwise positions in the wake are considered, and compared with the POD modes for the turbulent boundary layer without model. Compared with typical bluff-body flows, the eigenvalues convergence is slow, due to the high Reynolds number incoming turbulent boundary layer. For comparaison, only the first modes concentrate the energy for bluffs bodies (53 % of the total fluctuations energy in the two first modes for the square cylinder wake of Leite et al. (2018) for example). The eigenvalues are therefore comparable with previous works, in particular De Cillis et al. (2021): for a near-wake flow downstream a nacelle and tower model, the cumulative eigenvalues reached about 40 % after $N_m = 15$ modes, as it is about 50 % for the present study for the same number of modes.

Spatial POD modes obtained from the database are first presented here. It corresponds to the analysis performed in Raibaudo et al. (2022). In particular, maps of the spatial modes of the streamwise velocity component $\Phi_U$ for $x/D = 8.1$ and profiles along the spanwise $y$ direction and for two streamwise positions are presented in Fig. 6. Profiles are considered at the hub height ($z/D = 0.74$) and for different frequencies of surge-motion. Modes signs are adapted for a better comparison. A more complete analysis of the POD modes for the floating wind turbine can be found in Raibaudo et al. (2022). Two salient features can be observed from it: *(i)* the structure of the spatial modes forming around the disc shape are similar to previous works on POD analysis on wind turbines wakes (Bastine et al., 2015), and *(ii)* no significant effect of the surge-motion can be found in the spatial modes. Modes profiles are close in amplitude and shape, shown here for the streamwise velocity modes, but also for the other velocities components $v$ and $w$.

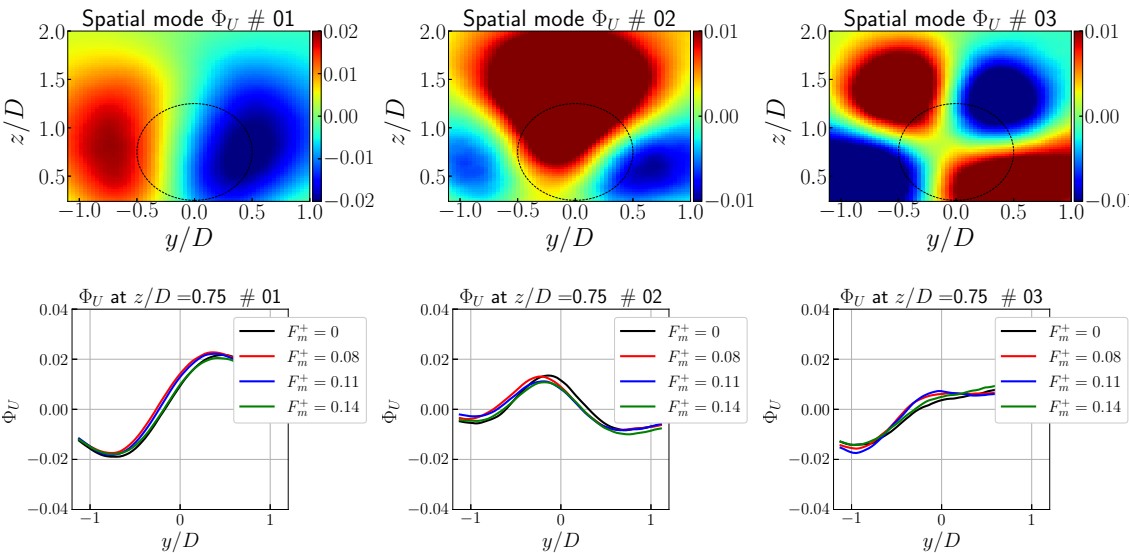

**Figure 6.** (a) Maps of spatial modes $\Phi_U$ for $x/D = 8.1$ for modes $i = 0$ to 2 and (b) profiles along the spanwise $y$ direction at the hub height $z/D = 0.74$ with the effect of the surge-motion. (Raibaudo et al., 2022).

However, significant differences can be observed on the temporal modes. Premultiplied PSD of the temporal modes $a_i$ are presented in Fig. 7 for the turbulent boundary layer without model and the two streamwise positions in the wake. Despite the low-sampling rate of the PIV system, the surge-motion imprint can be clearly identified.

A broad spectrum is found for the flow without model, centered around the main reduced frequency at $f^+ = 0.04$ ($f \approx 1$ Hz). By considering the fixed porous disc wake, the spectra amplitude changes significantly and the main frequency is shifted to $f^+ = 0.13$ ($f \approx 3$ Hz). The surge-motion is found to influence the temporal dynamics of the wake. The influence of the highest surge-motion frequencies is significantly visible: the spectrum is similar between a motion at frequency $F_m^+ = 0.08$ and the fixed model, where substantial peaks are observed for frequencies $F_m^+ = 0.11$ and $0.14$. The difference between low

and high frequencies is even more significant between modes 1 and 2: mode 1, especially in the far field, is mostly influence by the highest frequencies, where mode 1 is impacted by all the surge motion frequencies. As detailed after, it could be linked to a mitigation of respectively the horizontal meandering and the downwash phenomenon.

### 5.3   Physical interpretation of the modes dynamics

The relationship between the POD modes and the known physical phenomenon in wind turbine wakes is investigated by

considering the dynamics of the location of the wake center. The methodology corresponds to the one used by Bastine et al. (2015) and is used to estimate the center of the wake deficit. Using the deficit velocity $\tilde{\mathbf{u}}(\mathbf{x}, t) = U_0 - \mathbf{u}(\mathbf{x}, t)$, with $U_0$ the local freestream velocity, the center coordinates are calculated as:

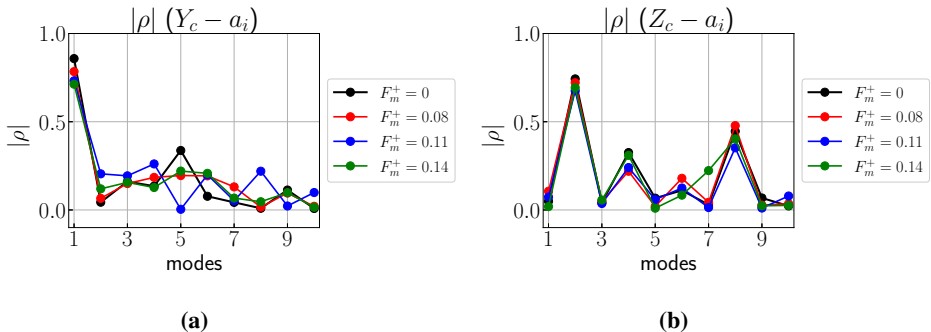

**Figure 7.** Premultiplied PSD of the temporal modes $a_i$ obtained from different configurations: (1) TBL without porous disc, (2) wake at $x/D = 4.6$ and (3) wake at $x/D = 8.1$; (a-d) modes 1 to 4.

**Figure 8.** Absolute value of the Pearson's correlation coefficients between the (a) spanwise and (b) wall-normal coordinates and the temporal POD modes.

$$\begin{bmatrix} y_c(t) \\ z_c(t) \end{bmatrix} = \frac{\iint \tilde{\mathbf{u}}^2(y,z,t) \begin{bmatrix} y \\ z \end{bmatrix} dy \, dz}{\iint \tilde{\mathbf{u}}^2(y,z,t) \, dy \, dz} \tag{6}$$

Correlations of these coordinates with the POD temporal modes are presented in Fig. 8. Pearson's correlation coefficient is considered here for the wake at $x/D = 8.1$. Conclusions are similar to the ones found by Bastine et al. (2015). The first mode strongly correlates with the spanwise motion of the wake, suggesting this mode corresponds to the horizontal meandering. The correlation decreases monotonically when the surge-motion frequency increases, from $\rho(y_c, a_1) = 0.86$ for the fixed model to $0.71$ for $F_m^+ = 0.14$. This decrease of the correlation with mode 1 is counterbalanced on the next three modes. The second mode correlates instead better with the wall-normal wake motion, which could correspond to the downwash flow motion and the vertical meandering. The correlation coefficient also decreases, from $\rho(z_c, a_2) = 0.74$ to $0.68$ for $F_m^+ = 0.11$. Contrary to $y_c$, correlation between the wall-normal position $z_c$ and the modes is maximum for modes 4 and 8. These higher modes are usually difficult to interpret as corresponding to a combination of motions. But for these modes 2, 4 and 8, the spatial modes (not shown here) consistency is mostly due to the wall-normal velocity component and a vertical motion of the wake. The global decreases of the correlation amplitude for the highest frequencies and observed for all the direction could then correspond to a full mitigation of the wake coherence and dynamics.

## 5.4 Reconstruction using multi-time delay LSE

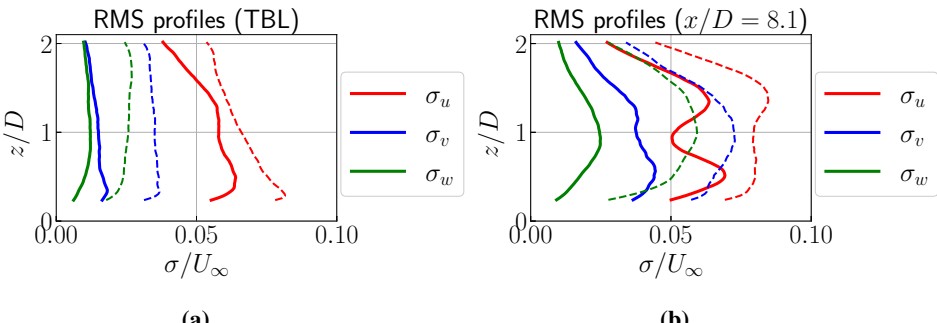

**(a)**    **(b)**

**Figure 9.** RMS reconstructed velocity $u/U_\infty$ profiles along $z$ using mtd-POD-LSE and original velocity using the same amount of modes $N_m = 100$. (a) TBL and (b) fixed model at $x/D = 8.1$. dashed lines: truncated original PIV data, lines: mtd-POD-LSE data

Following the methodology for reconstruction in section 5.1 and Eq. 4 in particular, the velocity fields are reconstructed at high sampling rate using the HWA time-resolved sensors. RMS velocity profiles of the reconstructed velocities along $z$ are shown in Fig. 9. A comparaison between the reconstructed and original velocities is therefore necessary to validate the reconstruction. As $N_m = 100$ POD modes are used to reconstruct the velocity, the comparaison is done not with the raw velocity profiles, but with the original velocity reconstructed with the same amount of modes.

For the turbulent boundary layer (Fig. 9(a)), the streamwise RMS velocity profile $\sigma_u/U_\infty$ is well reconstructed and reaches 60 % of the original velocity. The other reconstructed components $\sigma_v$ and $\sigma_w$ are underestimated (around 40 % for the spanwise and 50 % for the wall-normal component), but are still significant in amplitude. Same analysis can be made for the reconstruction of the wake flow (Fig. 9(b)). RMS of the reconstructed velocities are underestimated but show similar spatial distribution. Even if the stochastic estimation loses information similarly to the TBL, this underestimation could be linked to the conditioning of the correlation matrix during the POD process. The inversion of this matrix could amplify the smaller eigenvalues for ill conditioned LSE problems (Podvin et al., 2018). However, the difference between the projected and the reconstructed velocities are comparable to previous works using the same approach (Dekou et al., 2015; Podvin et al., 2018). Furthermore, the presence of the model generates a three-dimensional wake which is well captured by the PIV but not well enough by the one-dimensional hot-wires, which could also leads to errors.

Only the turbulent boundary layer and the far-wake for the fixed model are shown here, but the stochastic estimation was performed for the full database. By considering surge-motion, as its impact on the mean quantities is limited, the reconstructed profiles are similar than the ones presented in Fig. 9(b). For all the cases, the reconstructed velocities are reduced in amplitude by the POD-LSE process but conserves their main properties.

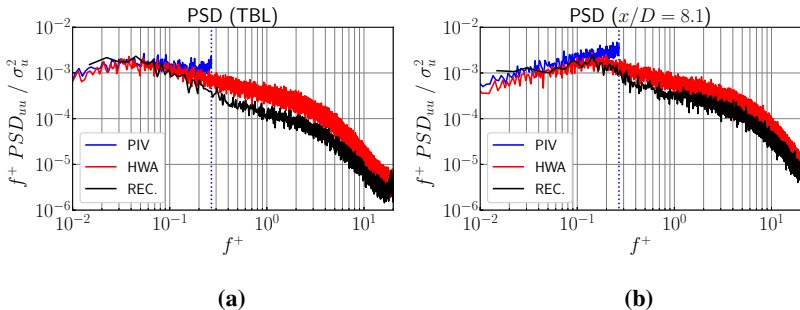

**(a)**  **(b)**

**Figure 10.** Premultiplied PSD of the reconstructed velocities at the hub height, compared to that directly measured by PIV or HWA. (a) TBL and (b) fixed model at $x/D = 8.1$. The vertical dashed line corresponds to the PIV sampling frequency limit.

Spectral analysis of the reconstructed streamwise velocity is presented in Fig. 10. Premultiplied PSD of the reconstructed velocity are compared with the ones estimated from the original PIV velocity truncated with the same amount of POD modes, and with a hot-wire sensor located at $y/D = 0$ and $z/D = 1.25$ (No. 10 on Fig. 1). The spectrum of the reconstructed velocity is consistent at low frequency with the PIV and follows well the dynamics behavior of the HWA sensor. A loss of spectrum amplitude is also observed between the two, suggesting a loss of information at higher frequency with the stochastic reconstruction as explained previously. Compared to the reconstructed turbulent boundary layer (Fig. 10(a)), the reconstructed wake shows peaks at low frequency around $f^+ = \mathcal{O}(10^{-1})$, which is consistent as expected with the spectral analysis performed on the temporal modes (Fig. 7).

## 6 FAST simulations using experimental velocities

Reconstructed velocity fields in the far field ($x/D = 8.1$) are now implemented as inflow wind conditions in FAST simulations, where the effect of the surge-motion and the flow dynamics on the computed produced power are considered in the present study.

### 6.1 Methodology

Spatial processing of the PIV velocity fields is first needed to adapt the velocity fields to the FAST requirements. As the
290 inflow wind conditions for FAST have to reach down the floor, the spatial POD modes are first extrapolated to be equal to zero at the wall. Velocity fields are then reconstructed with these extrapolated modes by POD projection (Eq. 1). The spatial range is then reduced to $Y_{FAST} = \pm 0.56\ D$ and $Z_{FAST} = [0, 1.38D]$ in a region close to the disc position and down to the wall. Interpolation is also required to have a square grid and an equal number of points ($N_{p,y} = N_{p,z} = 31$). Finally, the velocity fields are scaled to set the hub velocity $U_{hub,P} = 10$ m/s. Even if these operations are significant, features of the flow
are conserved, especially the reconstructed spatio-temporal dynamics of the flow obtained previously. All extrapolation and interpolation operations are performed using bivariate B-spline functions (Dierckx, 1981).

As a reference, generic turbulent velocity fields are also generated using TurbSim (Kelley and Jonkman, 2007), the turbulence generation tool provided with FAST. They correspond to velocity fields with prescribed boundary conditions and turbulence intensity, but without any coherence of their dynamics. For the boundary layer, the power law exponent is $\alpha = 0.15$, the surface
roughness length $z_r = 0.01$ m. Four turbulence intensities profiles are chosen for these fields: for the first three datasets, the turbulence intensities are set at the hub to 5 %, 7 % and 10 %. The last dataset follows the reference Normal Turbulence Model (NTM) in IEC 61400-1 requirements (International Electrotechnical Commission, 2006). Class C is chosen from this model, corresponding to the minimal turbulence class and a turbulence intensity of 12 % at the hub. The other parameters are the same as the fields from experiments ($U_{hub,P} = 10$ m/s, $N_{p,y} = N_{p,z} = 31$).

Simulations are performed using FAST as implemented in the software Qblade (v0.963), developed by TU Berlin (Marten et al., 2015). The tip-speed ratio (TSR) is fixed at 6.6, under the optimal value of 8 for this wind-turbine. The full-scale sampling frequency for the simulations and the aerodynamical model is $F_{s,FAST} = 100$ Hz. The simulation time is $T_{s,FAST} = 6400$ s, larger than the requirements of 10min, to ensure statistical convergence. Global and local variables, such as thrust, torque or strain gauges on the blades can be obtained. Only the power is presented here, the detailed analysis of the wind turbine
performance is left for future work.

### 6.2 Results

In order to demonstrate the benefit from using the proposed method, output rotor power obtained with the FAST simulations for different inlet flow configurations are presented in the current section. To ascertain the performances of the present approach against more standard techniques, output power of a single FOWT immersed in an atmospheric boundary layer are first
presented in Fig. 11 and compared to simulations performed using the TurbSim turbulent inflow generator available in FAST.

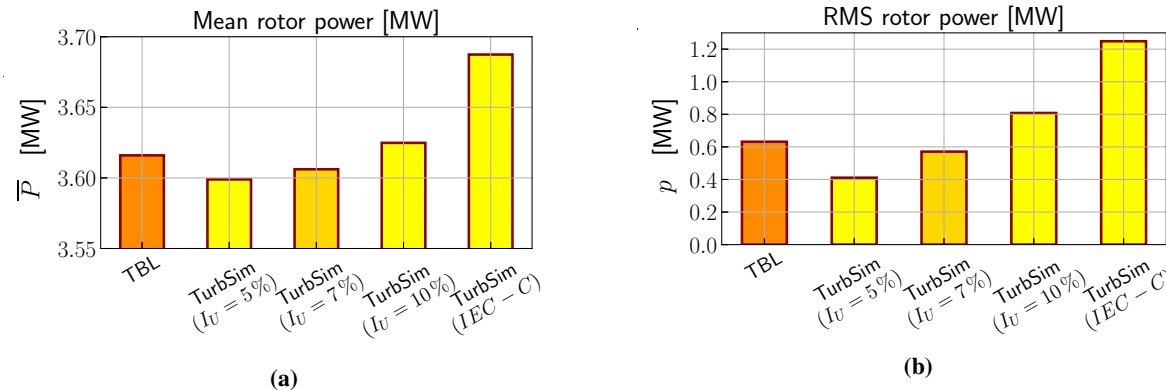

**(a)**            **(b)**

**Figure 11.** Comparison of the (a) mean and (b) RMS powers simulated with our experimental turbulent boundary layer (TBL) and the windfield datasets obtained with TurbSim, with four turbulence levels: $I_U = 5\%$, $7\%$, $10\%$ and reference Normal Turbulence Model in IEC 61400-1 requirements (class C).

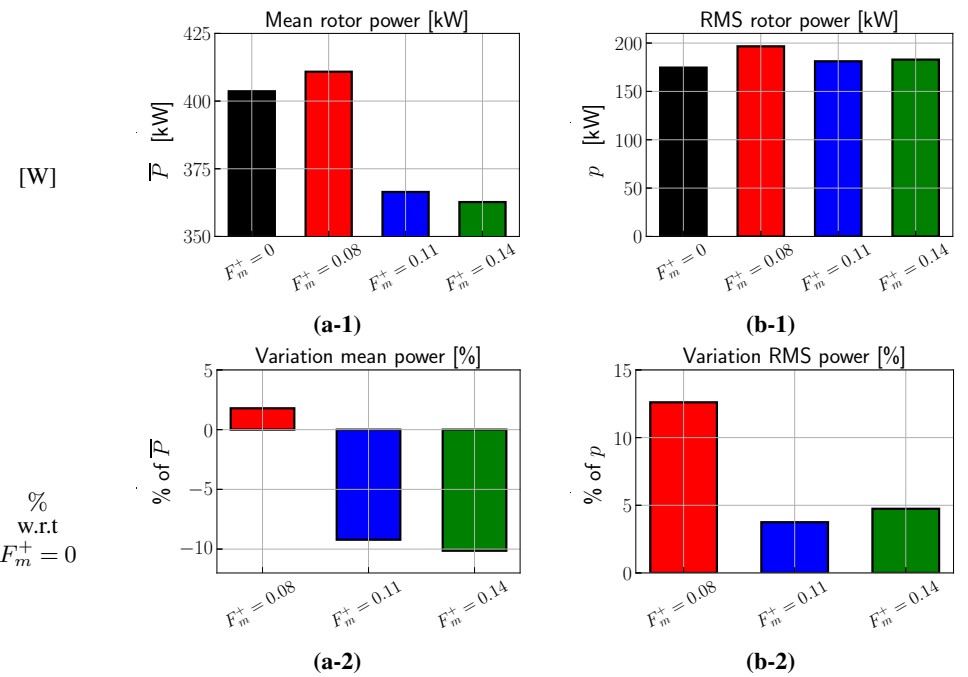

**Figure 12.** Effect of the wave-induced surge-motion on the powers obtained using Qblade / FAST simulations. (1) raw power; (2) variation in percent compared to the power without surge-motion. (a) mean rotor power for a second FOWT depending of the surge-motion frequency; (b) same but with RMS rotor power.

The RMS velocity for the reconstructed TBL is around 7 % at the hub, as presented in Fig. 9. The mean power matches well for simulations with inputs using our experimental fields and TurbSim (Fig. 11 (a)). A difference up to 0.3% is observed between these mean powers with comparable RMS velocity, in particular with $I_U = 7$ %. The RMS power obtained with experimental

fields also matches with the simulations with TurbSim datasets with similar turbulence levels. It should be noticed that the produced power is lower than 5MW due to the choice of the TSR.

In a second step, the ability of the present method to account for flow configurations more complex than the ABL alone is demonstrated by conducting FAST calculations of the performance of a wind turbine located in the wake of a first FOWT itself subjected to a surge-motion. Results are shown in Fig. 12. For a fixed upstream FOWT ($F_m^+ = 0.0$), the power produced for the second turbine decreased by 87 % compared to the unperturbed turbine. This difference is consistent with the power loss found in previous studies for an array of two turbines or inside farms (Ceccotti et al., 2016; Bartl et al., 2012; Corten et al., 2004; Eecen et al., 2006; El-Asha et al., 2017). For the lowest frequency tested ($F_m^+ = 0.08$), the mean power is similar to the fixed model (less than 2% of difference). For higher frequencies ($F_m^+ > 0.11$), the calculated power drops significantly, corresponding to a decrease of $-9.4\%$ and $-10.3\%$ for surge-motions frequencies of $F_m^+ = 0.11$ and $0.14$ respectively. RMS rotor power is shown to be particularly significant with respect to the mean power, approximately $50\%$ of $\overline{P}$ for all cases. An increase of the RMS power is shown for all surge motions compared to the fixed model. The threshold at $F_m^+ = 0.11$ observed for the mean power is also important for the fluctuating component: this increase of the RMS power is about 12 % for the lowest frequency, and $4 - 5$ % for higher frequencies.

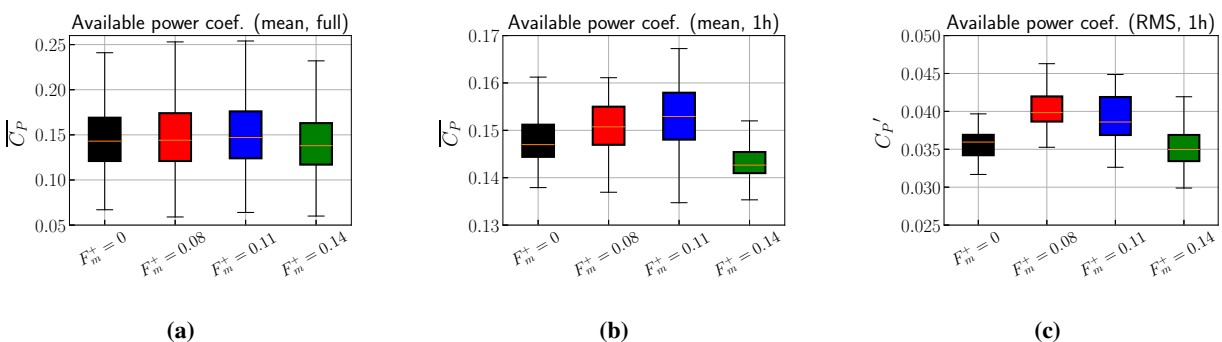

**Figure 13.** Statistics of the available normalized power computed directly from the inlet flow field. (a) Available power computed using the whole database; (b) and (c) mean and RMS, respectively, of the available power computed over 1 hour periods. Box plots show the median (white), first and third quartiles (main box) and minimum and maximum values (whiskers).

The attention of the reader is drawn to the fact that in this section, the FAST calculations are performed over a single full-scale period of one hour, which is not long enough to obtain statistically significant results and draw any definite conclusion on the performance of a wind turbine subjected to the wake of FOWT and the impact of the surge motion of the latter. To demonstrate this and the influence of the choice of the averaging period compared with the typical time scales of both the ABL and the FOWT wake, statistics of the available power (AP) directly computed from the PIV velocity fields are shown in Fig. 13. Figure 13(a) shows the distribution of the instantaneous AP estimated over the entire inflow database. Besides the fact that the range of variations is quite large, there is a clear trend in the median AP, with an increase with surge frequencies from 0 to $F_m^+ = 0.11$ and a drop for $F_m^+ = 0.14$. Distributions of the mean and RMS AP estimated over 1-hour periods are shown in Fig. 13(b) and (c), respectively. The range of variation of these short-period statistics can reach 25% of their median values.

It confirms the validity of the values estimated from the FAST calculations, which are within the range of variation of the statistics. It also shows that the first identified trends due the influence of the surge frequency (Fig. 12) are not representative and that a statistical analysis of the calculation output (which is beyond the scope of the present paper) is required to draw firm
conclusions.

## 7    Conclusions

A new method for generating inflow conditions for load simulations for wind turbines in complex configurations has been presented. It relies on interfacing the simulation code (FAST in the present case) with stereoscopic PIV wind-tunnel measurements performed in a cross-section of the flow. This new approach has been tested in various flow configurations, including an ABL
only but also the wake of an upstream wind turbine (modeled by a porous disc in the wind tunnel) submitted to a surge-motion, replicating the wave-induced movement a FOWT platform. The incoming motion was imposed with a fixed amplitude at four different frequencies.

The boundary layer, turbulence levels and large coherent structures of the turbulent flow upstream the model are taken into account in this study and are matching those of a realistic atmospheric flow. The wake was first characterized in a $y-z$ vertical
plane normal to the flow, at two streamwise positions via PIV and time-resolved hot-wire anemometry. By combining the two types of measurements, velocity fields were reconstructed at higher sampling frequency using multi time-delay POD-LSE. The temporal POD modes showed that the horizontal meandering and the vertical downwash are part of the main dynamics of the wake, and therefore included in the data. The reconstructed velocities were subsequently implemented in FAST as inflow conditions to estimate the performance (using FAST) of a full-scale 5MW wind turbine immersed in the wake of an upstream
wind turbine subjected to a surge motion. High surge frequencies of the upstream wind turbine lead to a significant reduction of the power produced by the downstream turbine. Here, a decrease up to 10.3 % of the power for the highest frequency is observed, which needs to be confirmed in future analyses.

Coupling experimental data from wind tunnel and numerical simulations for the full-scale turbine constitutes the originality of the approach presented here. The flow conditions (in particular the boundary layer and turbulence level) and the experimental
parameters (in particular the reconstructing sampling frequency) were chosen to match the requirements for FAST simulations. Extrapolations used to adapt the velocity fields as FAST inputs do not alter the mean and unsteady dynamics of the flow.

The present study focuses on a limited range of frequencies of motion in the streamwise direction. A possible improvement of the experimental set-up is the consideration of all 6 degrees-of-freedom for the motion and to increase the frequency range for better representativity. For the stochastic tool, classical regularization techniques could help to improve the reconstruction
quality. For FAST simulations, it could be interesting to consider floating wind turbines for the model to understand the interaction between the experimental inputs and the FOWT performances simulated by FAST.

As the spatial modes are unchanged (Fig. 6) and the physical differences are mainly contained in the temporal modes, representative velocity fields can be easily generated using a reduced set of parameters. From a low-order modeling strategy, it means that a unique set of spatial modes could be used, combined with artificial temporal modes having the proper dynamics

in order to generate flow fields corresponding to generic floating wind turbines. It could allow for example real-time estimation of the wind turbine performances based on limited measurements on the full-scale prototype and *a priori* information from wind tunnel data.

*Author contributions.* L.P. conceived of the presented idea, L.P. conceived and designed the experiments, C.R. and L.P. carried out the experiments and collected the data, C.R. and J.-C.G. designed and analysed the FAST simulations, C.R. wrote the basic version of the paper, 380 C.R. and L.P. wrote the final version of the manuscript, L.P. supervised the project and got the funding. All authors provided critical feedback and helped shape the research, analysis and manuscript.

*Competing interests.* The authors declare that they have no known competing interests or personal relationships that could have appeared to influence the work reported in this paper.

*Acknowledgements.* The authors acknowledge the financial support of WEAMEC, the West Atlantic Marine Energy Community, under the 385 grant N° 2020 WIND2SIM. The authors are indebted to T. Piquet, research engineer at LHEEA, for his help during the experiments.

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
