# Peer review of "Realistic turbulent inflow conditions for estimating the performances of a floating wind turbine"

_Wind Energy Science, 2023_

## Referee Comment (RC1)

**WES 2023 paper review: Realistic turbulent inflow conditions for estimating the performances of a floating wind turbine**

from Cédric Raibaudo, Jean-Christophe Gilloteaux, and Laurent Perret

July 10, 2023

**General comments**

The present paper proposes a method to generate realistic turbulent inflow for floating wind turbine simulations, where the inflow generated is corresponding to the far-wake behind a floating turbine, around 8D downstream. The paper is overall well written, clear, short but explains sufficiently the aim of the study and the content is quite convincing. The work done is novel as it combines experimental approach to generate realistic inflow conditions for floating wind simulations based on stochastic method. The paper is well suited to be published in the Wind Energy Science journal. However, certain aspects should be revised and some parts be enhanced in order to improve the quality of the manuscript. These are detailed hereafter.

**Specific comments**

**Literature review and motivation**

The review of the literature and motivation of the study are well presented, showing the need to better account for wake effects of floating wind turbine in aero-elastic simulations. However, some recent studies are omitted in the citation and would add more value to the literature review to show the importance of floating wind turbine's wake dynamic. Among others:

- Chen, G., Liang, X. F., Li, X. B. (2022). Modelling of wake dynamics and instabilities of a floating horizontal-axis wind turbine under surge motion. Energy, 239, 122110.

- Li, Z., Dong, G., Yang, X. (2022). Onset of wake meandering for a floating offshore wind turbine under side-to-side motion. Journal of Fluid Mechanics, 934, A29.

- Messmer, T., Hölling, M., Peinke, J. (2023). Enhanced recovery and non-linear dynamics in the wake of a model floating offshore wind turbine submitted to side-to-side and fore-aft motion. arXiv preprint arXiv:2305.12247.

These three papers show numerically and experimentally the impact of floating wind tubines' motions on wake recovery and dynamics.

The paragraph which starts at line 83 to 94 could be revised. As discussed after in this paper review, do you really account for wave-induced surge motion in your experiments? I would suggest to write more general about surge motion. Please see "technical corrections" for some improvements.

**2. Experimental set-up, 3. Mean structure of the flow, 4. Scaling**

Overall these three sections are well presented and important for the reader. The experimental set-up is well described and enough information are given to enable reproduction. There is, however, some confusion with the inflow conditions which should be clarified. In line 102, you write: "without model, the free-stream velocity is $U_\infty = 4.2$ m/s and the turbulent intensity about 0.5%". The inflow used for the experiments is, if well understood, an atmospheric boundary layer with a mean TI of 8 % (as said after in the paper and to be seen in the figure 3). So you should avoid such a sentence that confuses more the reader and does not help for the understanding. Also, even if it is cited, the characteristic of the inflow (profile of TI and estimation of the integral length scale) could be specified.

Regarding the disk used, the $C_t$ value seems quite low in comparison to real wind turbine operation where $C_t$ is usually in the order of $0.8 - 0.9$. Why did you choose such a low $C_t$? This surely impacts the development of the wake and represents above rated conditions that are less of interest in my point of view. A few sentences on this choice would be appreciable.

In the section "3. Mean structure of the flow", you describe the mean flow in the wake at 4.6D and 8.1D. You also refer to previous studies of yours on the same set-up. You could discuss more the wake in terms of its development, is the wake at 8.1D the fully developed far-wake or is it still in the transition region. Regarding the high inflow TI, you might at 8.1D already be in the far-wake but the low $C_t$ might shift the region where the far-wake starts. I specifically make this comment because on your TI profiles in the y-axis from your previous paper, we clearly see the two peaks in the TI at x = 4.6D as well as at x = 8.1D, which are characteristic of the so-called shear layers that have not merge yet. Also what is the amount of recovery at 8.1D (and power available), this is then important when you compare the power produced by a turbine placed in this region.

The part about the scaling explains well the comparison between the model to a real 2MW or 5MW turbine. The last paragraph (lines 156 to 159) could be re-written. As you write, the wave-induced motion are happening at a frequency that is too high to be reproduced experimentally here. In order of magnitude, for a wind speed of 10 m/s, wave-induced motion gives $F_m^+$ of 0.8-1.0 which is above the range you investigated. In your case, it seems you cover more the low frequency-large amplitude types of motion (i.e driven by the moorings). You could be clearer on that, especially because previous studies of Chen et. al (2022), Li et al. (2022) and Messmer at al. (2023) showed how the wake dynamics depend greatly on $F_m^+$. So in this study, you cover a specific range of motions (around the surge natural frequency) but not all types of FOWT's motion.

I would suggest that you reorganise the three sections, you could as you did, start by the experimental set-up then write about the scaling and then detail the cases you investigated.

**5. Stochastic reconstruction using POD-LSE multi-delays**

The methodology used to reconstruct the velocity fields could be more detailed, so that the reader do not need to read other references to understand your approach. For instance, how do you exactly determine $a_i$, this is only partially described and could be enhanced.

From the results presented in figure 4, should we conclude that 100 modes are enough to resolve 80 % of the flow? If not, you could detail what do the eigenvalues represent.

It is good that you show the spatial modes (figure A.1). I think you could even include them in the paper, since they describe some important structure in the flow. You could also comment more about the physical meaning of these structures. As you write, it looks like mode 1 is associated to lateral meandering wherease mode 2 could be some pulsing mode and mode 3 a combination. What is your interpretation on that?

The results of figure 5 are very interesting and could be more interpreted. Is there any reason why you plot pre-multiplied spectra rather than the normal spectra? If yes, you could explain why. Panel (a-3) shows that the mode 1 is excited by the movements for the higher frequencies ($F_m^+ = 0.11$-$0.14$) but not the lowest. In contrast mode 2 is excited by the three frequencies. What is your interpretation on that?

You mention that the wake of the fixed disk shows high-energy peak at $f^+ \approx 0.13$, which is seen in the psd of the time coefficient of mode 1 but not mode 2, this is most certainly the natural meandering frequency of the disk. This might also explain why you see some peak for $F_m^+ = 0.11$-$0.14$, very close to $f^+ \approx 0.13$. With respect to real floating wind turbines, it would be interesting to see in which range the natural frequency of meandering is observed. From my experience and from previous work, it is in the range $f^+ \approx 0.1 - 0.4$.

In the sub-section 5.3 "physical interpretation of the modes dynamics", you show some correlations between the time coefficients and the position of the center of the wake but do not really interpret these results. I wonder what the lowest correlation for $F_m^+$ for mode 1 between the time coefficient and the wake center really mean? If you would do this analysis with the wake edges, i.e in the shear region which separates the inner wake to the incoming flow, perhaps the results would be quite different and you might observe a higher correlation with mode 1 for $F_m^+ \approx 0.11 - 0.14$ showing that these frequencies excite even mode meandering.

In the sub-section 5.4 "reconstruction using multi-time delay LSE", it is to me not clear how you reconstruct the wind field based on the data you have. Please update this section to make it easier to understand. As you write, the method you used to reconstruct the wind field lead to a quite large loss of information. In fact, as Figures 7 and 8 show, the reconstructed field is less energetic than the original field, particularly for $f^+ > 0.25$, which may explain why the reconstructed RMS profiles are smaller in magnitude. Do you have any ideas how to modify your reconstruction method in order to better account for these smaller structures in the flow? You then use this reconstructed flow for the simulations, how do you think this less resolved flow impact your results? You should comment on that in the paper.

**6. Implementation for FAST simulations**

This section is about the utilisation of the reconstructed wind field for FAST simulations, which is a central part in the paper. Does the turbine placed in the flow at 8D is fixed or also enabled to move, you should mention it.

I find good that you show some reference simulations in figure 9, which show that for a given mean wind speed, the higher the TI, the higher the power produced. I interpret this as following: the higher the inflow TI, the larger the fluctuations, and to the power three, the higher fluctuations include more power than for the lowest TI which is extracted by the turbine and does explain why power produced increase with an increase of TI. Your simulations are done below optimum ($TSR \approx 6.6$), is there any particular reason for that?

How realistic is the final interpolated flow with respect to a real wake flow in offshore conditions? Did you have enough time of experiments to reproduce 6400 s at full scale how do the data repeat throughout a reconstructed wind field?

When you present the results of the power produced by the turbine in the wake, you write that the power is about 87 % less than the unperturbed turbine. Could you comment this with respect to the amount of recovery of the wake at 8.1D (this goes with my above comments that you should estimated the amount of power available in the wake, somehow equivalent to the amount of recovery). So from the figure 3, if the mean wind speed in the wake is 0.55 $U_\infty$, then the power available is $\sim 0.17 U_\infty^3$, which is in line with your results. It is however really a standard value at 8D downstream of a turbine? Don't we except at this location more recovery and thus more power available for a downstream turbine? In Porté-Agel et al (2020), Figure 13 shows the power produced by a wind farm on the different rows and we can see that the second row produces around 0.6 of the power of the first row, which I think is more realistic. This suggests that the cases you examined may not be the closest to reality, do you agree?

You then show differences in the power produced between the cases (turbine placed in the wake of the fixed and the surging turbine). I wonder where the differences you observed come from. Are you measurements enough resolved that you can ensure these results are outside of the uncertainty range? Comparing the power available in the flow for each of these cases could help to explain the results. Maybe a look at the mean TI in the wake as well. How do the approximations of the reconstructed field play a role in these results?

**Conclusion**

The conclusion is a good summary of your work. I would however be careful with the strong conclusion that in the wake of the moving turbine, you have up to 10 % less power produced by a turbine placed in it. This is in the context of your study but might not be a general statement.

**Technical corrections**

The following are some technical corrections that could be made. Overall, many sentences should be checked and rewritten if necessary, and particular attention should be paid to the tenses used.

- Abstract: the scale 1:750 is with respect to a 5MW wind turbine but your set-up is a proper scale-down of a 2MW system. Before writing about surge motion, you could introduce the topic of floating wind turbine and the extra 6-DoF.

- line 16: "in recent years" or "over the last years" instead of "for the recent years".

- line 18: you could add a reference to this statement

- line 28: sentence to be reviewed

- line 32: maybe one reference about wake region would be appreciable (Porté-Agel et al. (2020), Vermeer et al. (2003) or Neunaber et al. (2020)).

- line 36: sentence to be reviewed

- line 90-94: I would suggest to avoid writing "will be presented", in a paper I would rather write "is presented". Then "the velocity fields are reconstructed..." instead of "need to be.."

- Figure 1: the letters (a) and (b) could be centred, the image could be bigger and vectorised scheme for panel (b) could be used, i.e which conserves the quality when zooming in.

- line 102: sentence to be removed

- line 138: the chosen prototype*

- Figure 5. why do you present results for mode 4 if you don't show it and don't write about it?

- lines 206-207: to be corrected

- lines 223-225: make this sentence more clear

- Figure 7. you should adapt the line style so that we clearly see the differences between the original profiles and the reconstructed profiles and change the legend.

- line 226: reaches 60 % of what??

- line 236: "are shown here" not "were shown".

- line 248: change the title

---

## Author Comment (AC1)

**Realistic turbulent inflow conditions for estimating the performances of a floating wind turbine**

C. Raibaudo, J.-C. Gilloteaux, L. Perret

We thank the reviewers for their valuable comments and suggestions. We have addressed all comments and incorporated most suggestions, except when these conflicted between reviewers. Major modifications and additions to the text have been highlighted in blue in the manuscript for better readability.

A rebuttal to the individual reviewer is included below. The number used to identify the reviewer is as provide to us by WES.

**Reviewer #1**

**Literature review and motivations**

1. The review of the literature and motivation of the study are well presented, showing the need to better account for wake effects of floating wind turbine in aero-elastic simulations. However, some recent studies are omitted in the citation and would add more value to the literature review to show the importance of floating wind turbine's wake dynamic.

R1: The two first references have been added to the literature section. They are indeed particularly interesting and have been missed to their recent publication. The excellent work of T. Messmer, who I had the opportunity to meet at the TORQUE and WESC conferences, is also cited via the *Journal of Physics:Conference series* article. But I am uncomfortable citing an arXiv-paper which has not been peer-reviewed yet.

2. The paragraph which starts at line 83 to 94 could be revised. [...] Do you really account for wave-induced surge motion in your experiments? I would suggest to write more general about surge motion.

R2: The paragraph has been rephrased, to focus on the methodology of coupling experimental data from wind tunnel experiment and FAST simulations. The surge motion is also discussed in this paper, but it was the original intention of the authors to focus on the methodology. A more detailed analysis of the surge-motion effect on the WT wake will be performed in a future article.

**Experimental set-up, mean structures and scaling**

3. In line 102, you write: "*without model, the free-stream velocity is $U_\infty = 4.2$ m/s and the turbulent intensity about 0.5%*". The inflow used for the experiments is, if well understood, an atmospheric boundary layer with a mean TI of 8 % (as said after in the paper and to be seen in the figure 3). So you should avoid such a sentence that confuses more the reader and does not help for the understanding.

R3: The sentence has been precised and moved to the beginning of the paragraph. The RMS and not the TI is plotted in figure 3, so a RMS value of 8 % leading to a TI of 13 %. The turbulence level of the free-stream flow without model is however important to precise that the TI of 13 % is mostly due to the turbulent boundary layer development.

4. The characteristic of the inflow (profile of TI and estimation of the integral length scale) could be specified.

R4: A sentence of the integral length scale has been added. Scaled for the full scale wind turbine, it corresponds to an integral length scale about 200 m, that fits the requirements of the VDI guideline 3783 VDI [9]. The flow without

model is well documented in previous works (Raibaudo and Perret [5], Schliffke et al. [7]) and the authors did not want to include any mean and RMS profile of the turbulent boundary layer for concision. However, a figure and a paragraph have been added on the turbulent flow without model, following the work of Raibaudo and Perret [5].

5. Regarding the disk used, the $C_t$ value seems quite low in comparison to real wind turbine operation where $C_t$ is usually in the order of 0.8 - 0.9. Why did you choose such a low $C_t$?

R5: The disc used for the present study is the same as the one used by Schliffke et al. [7]. A first justification of the use of this disc is a deepening of the documentation of this wake in a different spatial plane and steroscopic PIV. As detailed by Schliffke et al. [7], the configuration is matching well with the flow and wind turbine properties of the 2MW FOWT FLOATGEN research project, installed at Centrale Nantes' offshore test site in Le Croisic, France (Rousset et al. [6]). The objective was therefore to have a surrogate of the real size wind turbine capable to be tested in wind tunnel.

6. In the section "3. Mean structure of the flow", you describe the mean flow in the wake at 4.6D and 8.1D. You also refer to previous studies of yours on the same set-up. You could discuss more the wake in terms of its development, is the wake at 8.1D the fully developed far-wake or is it still in the transition region. Regarding the high inflow TI, you might at 8.1D already be in the far-wake but the low Ct might shift the region where the far-wake starts. I specifically make this comment because on your TI profiles in the y-axis from your previous paper, we clearly see the two peaks in the TI at x = 4.6D as well as at x = 8.1D, which are characteristic of the so-called shear layers that have not merge yet. Also what is the amount of recovery at 8.1D (and power available), this is then important when you compare the power produced by a turbine placed in this region.

R6: We thank the reviewer for his helpful comment. The visible extrema in the rms spanwise profiles at hub height indicate that the wake has not fully recovered (we would otherwise expect a single maximum). However, at $z = z_{hub}$ and $x = 8.1D$, the mean velocity $u$ reaches 60 % of the undisturbed velocity at the same height, from $4.6D$ to $8.1D$, the TI clearly decays, indicating that the $8.1D$ location is in the transition region towards the far wake.

The following sentence has been added at the end of Section 4.

*"It must be noted here that, at the most downstream location $x/D = 8.1$, two distinct peaks in the horizontal profile of the RMS of the streamwise velocity component measured at hub height are still visible, indicating that the wake is not fully developed yet. However, the recovery of the mean velocity $u(z = z_{hub})/U_{hub}$ reaches 60 % of the undisturbed velocity, with a corresponding decreasing turbulent intensity. The wake is therefore still in its transition phase towards a fully developed wake."*

As a final note, we agree with the reviewer that documenting the wake development is important. In the present case, the flow has been investigated in details by our group, in particular by Schliffke and co-workers. In the proposed paper, we limited the analysis of the flow to a minimum as the central point of the paper is the presentation of a novel methodology for generating turbulent inflow conditions and its potential application to the study of wind turbine performance when immersed in complex turbulent flow.

7. The part about the scaling explains well the comparison between the model to a real 2MW or 5MW turbine. The last paragraph (lines 156 to 159) could be re-written. [...] In order of magnitude, for a wind speed of 10 m/s, wave-induced motion gives $F_m^+$ of 0.8 - 1.0 which is above the range you investigated. In your case, it seems you cover more the low frequency-large amplitude types of motion (i.e driven by the moorings).

R7: the paragraph has been rephrased. The displacement system has been designed to match the behavior of the SEM-REV test site where the FLOATGEN prototype of diameter $D = 80$ m is installed. Schliffke et al. [7], Tarpin [8]. The probability chart for this test site site is plotted in figure 1. The most probable characteristic wave period is $t_p = 11$ s, leading to a wave-induced motion of $F_m^+ = 0.7$, which is indeed low but of the order of magnitude of the imposed model motion. But part of the operating range still matches the imposed motion frequency and the order of magnitude is still the same. By taking into account realistic boundary layer profiles, turbulence levels, these moderate surge-motions frequencies and experimental approach, the representativity of this study is still significant compared to previous works.

8. I would suggest that you reorganise the three sections, you could as you did, start by the experimental set-up then write about the scaling and then detail the cases you investigated.

R8: the order of sections has been reorganised.

**Stochastic reconstruction**

[Figure]

**Fig. 1** Probability chart of sea states at the SEM-REV test site. From Schliffke et al. [7] and based on a work of Tarpin [8]

9. The methodology used to reconstruct the velocity fields could be more detailed, so that the reader do not need to read other references to understand your approach. For instance, how do you exactly determine $a_i$, this is only partially described and could be enhanced.

R9: even if the stochastic methodology is standard in fluid mechanics and has been detailed in previous conference papers (in particular in Raibaudo and Perret [5]), details have been added to the paper concerning the POD-mLSE reconstruction.

10. From the results presented in figure 4, should we conclude that 100 modes are enough to resolve 80 % of the flow?

R10: Absolutely. Besides, this affirmation was precised in line 171.

11. It is good that you show the spatial modes (figure A.1). I think you could even include them in the paper, since they describe some important structure in the flow. You could also comment more about the physical meaning of these structures. As you write, it looks like mode 1 is associated to lateral meandering whereas mode 2 could be some pulsing mode and mode 3 a combination. What is your interpretation on that?

R11: the appendix has been moved to the main text, and the explanations have been rephrased. The spatial modes interpretation alone is still a difficult task and this is why it was more rigorous to study also the temporal modes behavior. Coupled with the correlation between the coordinates of the wake center and the temporal modes (Fig. 6), the mode 2 is more probably the effect of the vertical meandering, the downwash phenomenon and the turbulent boundary layer interaction with the model.

12. Is there any reason why you plot pre-multiplied spectra rather than the normal spectra? If yes, you could explain why.

R12: in addition to compare easily with previous works (Schliffke et al. [7], the pre-multiplied has been preferred in the turbulence community for multiple reasons, one is that the integral of the curve of the pre-multiplied PSD is equal to the turbulent kinetic energy. Another argument is to be able to find a $k^{-1}$ plateau (called the Kolmogorov plateau) in the inertial region.

13. Panel (a-3) shows that the mode 1 is excited by the movements for the higher frequencies ($F_m^+ = 0.11 - 0.14$) but not the lowest. In contrast mode 2 is excited by the three frequencies. What is your interpretation on that?

R13: Sentences have been added to precise this very interesting point. A possible interpretation is linked to the physical analysis performed in the next section: mode 1 corresponds to the horizontal meandering, where mode 2 corresponds to the vertical meandering and the downwash phenomenon. Mode 1 seems to react better to the high-frequencies surge motion, especially when these frequencies reach what is thought to be the natural meandering frequency. Mode 2 however corresponds to the downwash vertical motion of the wake, which seems to be mitigated whatever the frequency considered.

14. With respect to real floating wind turbines, it would be interesting to see in which range the natural frequency of meandering is observed. From my experience and from previous work, it is in the range $f^+ \approx 0.1$ - $0.4$.

R14: the natural frequency of the prototype we wanted to replicate (the FLOATGEN prototype) is not accessible. But the previous works in wind tunnel and few onsite works (especially the recent paper of Li. et al. (2022) you suggested) seems to confirm the order of magnitude you proposed.

15. In the sub-section 5.3 "physical interpretation of the modes dynamics", you show some correlations between the time coefficients and the position of the center of the wake but do not really interpret these results. I wonder what the lowest correlation for $F_m^+$ for mode 1 between the time coefficient and the wake center really mean?

R15: A sentence has been added to comment on the global decrease of the correlation amplitude. For us, it means a dynamical phenomenon that is less coherent, in amplitude and phase, which could corresponds to a full mitigation of the meandering and the downwash.

16. In the sub-section 5.4 "reconstruction using multi-time delay LSE", it is to me not clear how you reconstruct the wind field based on the data you have. Please update this section to make it easier to understand.

R16: A sentence referring to the section Methodology 5.1 has been added. The principle is mostly related to equation (2) to reconstruct the temporal modes from time-resolved sensors.

17. In fact, as Figures 7 and 8 show, the reconstructed field is less energetic than the original field, particularly for $f^+ > 0.25$, which may explain why the reconstructed RMS profiles are smaller in magnitude. Do you have any ideas how to modify your reconstruction method in order to better account for these smaller structures in the flow? You then use this reconstructed flow for the simulations, how do you think this less resolved flow impact your results? You should comment on that in the paper.

R17: as explained in the paper (lines 229 to 235), a decrease of the reconstructed amplitude is usual for this type of analysis. It is usually due to the conditioning of the correlation matrix that is the kernel of the decomposition. We used SVD instead of classical inversion techniques to invert the kernel. A possible improvement is to use regularisation, like Tikhonov regularization, which has been proved to be efficient with similar stochastic tools applied for fluid mechanics fields.

**Implementation for FAST simulations**

18. This section is about the utilisation of the reconstructed wind field for FAST simulations, which is a central part in the paper. Does the turbine placed in the flow at 8D is fixed or also enabled to move, you should mention it.

R18: In FAST, the turbine is fixed but see FOWT wakes as inputs. The implementation of FOWT in Qblade has been performed long after the end of the research project. This consideration would be a perspective for future works.

19. Your simulations are done below optimum ($TSR \approx 6.6$), is there any particular reason for that?

R19: this non-optimal value of TSR has been chosen for consistency with previous (unpublished) work performed in our group. It was originally based on the link between the thrust coefficient and the tip-speed-ratio (expressed for example by Eltayesh et al. [4]).

20. How realistic is the final interpolated flow with respect to a real wake flow in offshore conditions? Did you have enough time of experiments to reproduce 6400 s at full scale how do the data repeat throughout a reconstructed wind field?

R20: by respecting the time scaling factor $\Lambda_t$ explicited in section 4 and explained also in the methodology section 6.1, the simulation time for the full scale of $T_{s,FAST} = 6400\,s = 107\,min$ corresponds to 23 $s$ of experimental data with respect to $\Lambda_t$. But this is much larger than the recommended 10 $min$ from IEC. As the paper is a first explanation of the coupling methodology between the experiments and the FAST simulations, the authors believe this is sufficient to present the method. But a more detailed study will be done based on the last results from figure 11 and the sensitivity of the incoming flow inputs on the FAST results.

21. When you present the results of the power produced by the turbine in the wake, you write that the power is about 87 % less than the unperturbed turbine. Could you comment this with respect to the amount of recovery of the wake at 8.1D (this goes with my above comments that you should estimated the amount of power available in the wake, somehow equivalent to the amount of recovery). So from the figure 3, if the mean wind speed in the wake is 0.55 $U_\infty$, then the power available is $0.17 U_\infty^3$, which is in line with your results. It is however really a standard value at 8D downstream of a turbine? Don't we except at this location more recovery and thus more power available for a downstream turbine?

R21: the value of 87 % is indeed high, but not far from the ones found in the literature from wind tunnel, LIDAR measurements and simulations, between 60 % and 80 % (Corten et al. [1], Eecen et al. [2], El-Asha et al. [3]). These references have been added to the paper.

22. Figure 13 shows the power produced by a wind farm on the different rows and we can see that the second row produces around 0.6 of the power of the first row, which I think is more realistic. This suggests that the cases you examined may not be the closest to reality, do you agree?

R22: As explained in the previous comment, the range found in the literature for power losses is between 60% and 80 %. The power loss found here is indeed high, but not inconsistent with this range.

23. You then show differences in the power produced between the cases (turbine placed in the wake of the fixed and the surging turbine). I wonder where the differences you observed come from. Are you measurements enough resolved that you can ensure these results are outside of the uncertainty range?

R23: as explained in the scaling section and the methodology for FAST simulation subsection, the resolution in time and the period used here fits well with the requirements for this type of simulations for the full scale, respectively 20 Hz for the aerodynamic simulation and 10min minimum of simulation.

24. Comparing the power available in the flow for each of these cases could help to explain the results. Maybe a look at the mean TI in the wake as well. How do the approximations of the reconstructed field play a role in these results?

R24: the results shown in figure 13 are based on the raw PIV data only. Hence there is no reconstruction involved here. The objective was to obtain an explanation of the surge influence observed with the FAST simulation in Fig. 12, removing the possible impact of the reconstruction method. Figure 13 shows that the surge motion has a direct influence on the available power. The physics of this is currently under investigation.

25. I would however be careful with the strong conclusion that in the wake of the moving turbine, you have up to 10 % less power produced by a turbine placed in it. This is in the context of your study but might not be a general statement.

R25: this sentence has been rephrased with precautions.

**Technical corrections**

26. Abstract: the scale 1:750 is with respect to a 5MW wind turbine but your set-up is a proper scale-down of a 2MW system. Before writing about surge motion, you could introduce the topic of floating wind turbine and the extra 6-DoF.

R26: two sentences have been added in the abstract.

27. line 16: "in recent years" or "over the last years" instead of "for the recent years".

R27: It has been replaced, thank you.

28. line 18: you could add a reference to this statement

R28: A reference has been added.

29. line 28: sentence to be reviewed

R29: the sentence has been rephrased and simplified.

31. line 32: maybe one reference about wake region would be appreciable (Porté-Agel et al. (2020), Vermeer et al. (2003) or Neunaber et al. (2020)).

R31: line 32 is dedicated to the work of Aubrun et al. (2013), completed by the study of Camp and Cal (2016) on the definition of regions in the wake. One reference was therefore added at the end of the paragraph.

32. line 36: sentence to be reviewed

R32: The sentence has been simplified and cut in two.

33. line 90-94: I would suggest to avoid writing "will be presented", in a paper I would rather write "is presented". Then "the velocity fields are reconstructed..." instead of "need to be..".

R33: the sentences have been rephrased.

34. Figure 1: the letters (a) and (b) could be centred, the image could be bigger and vectorised scheme for panel (b) could be used, i.e which conserves the quality when zooming in.

R34: figures (a) and (b) have been replaced by their vectorised versions, they have been enlarged and the letters centered.

35. line 102: sentence to be removed
R35: whatever sentence is concerned by this comment (the calculus of the freestream velocity + turbulence intensity *or* the boundary layer height + the Reynolds number), the authors believe these informations are usefull to define the incoming flow without model. The sentence has been rephrased, as explained in comment n° 3.

36. line 138: the chosen prototype*
R36: the typo has been corrected, thanks.

37. Figure 5. why do you present results for mode 4 if you don't show it and don't write about it?
R37: a sentence for mode 4 has been added. The POD modes are usually analysed by pairs, and mode 4 is here to ensure the comparaison and possible pairing with mode 2 or/and mode 3.

38. lines 206-207: to be corrected
R38: these lines have been precised. As explained in comment n° 13, the full paragraph has also been improved.

39. lines 223-225: make this sentence more clear
R39: This sentence has been precised. To compare the results with the reconstructed original velocity and not with the raw data is usual for this type of stochastic tools.

40. Figure 7. you should adapt the line style so that we clearly see the differences between the original profiles and the reconstructed profiles and change the legend.
R40: Figure 7 and its legend have been changed.

41. line 226: reaches 60 % of what??
R41: this has been rephrased.

42. line 236: "are shown here" not "were shown".
R42: this has been corrected.

43. line 248: change the title
R43: the title has been changed.

**References**

[1] Gp Corten, P Schaak, and T Hegberg. Velocity profiles measured above a scaled wind farm. Energy research Centre of the Netherlands, (November):22–25, 2004. URL `ftp://ftp.ecn.nl/pub/www/library/report/2004/rx04123.pdf`.

[2] P. J. Eecen, S. A.M. Barhorst, H. Braam, A. P.W.M. Curvers, H. Korterink, L. A.H. Machielse, R. J. Nijdam, L. W.M.M. Rademakers, J. P. Verhoef, P. A. Vd Werff, E. J. Werkhoven, and D. H. Van Dok. Measurements at the ECN Wind Turbine Test location Wieringermeer. In European Wind Energy Conference and Exhibition 2006, EWEC 2006, volume 2, pages 1477–1480, 2006. ISBN 9781622764679.

[3] Said El-Asha, Lu Zhan, and Giacomo Valerio Iungo. Quantification of power losses due to wind turbine wake interactions through SCADA, meteorological and wind LiDAR data. Wind Energy, 20(11):1823–1839, 2017. ISSN 10991824. doi: 10.1002/we.2123.

[4] Abdelgalil Eltayesh, Magdy Bassily Hanna, Francesco Castellani, A. S. Huzayyin, Hesham M. El-Batsh, Massimiliano Burlando, and Matteo Becchetti. Effect of wind tunnel blockage on the performance of a horizontal axis wind turbine with different blade number. Energies, 12(10):1–15, 2019. ISSN 19961073. doi: 10.3390/en12101988.

[5] Cédric Raibaudo and Laurent Perret. Low-Order Representation of the Wake Dynamics of Offshore Floating Wind Turbines. 12th International Symposium on Turbulence and Shear Flow Phenomena, TSFP 2022, pages 1–6, 2022.

[6] Jean-Marc Rousset, Hakim Mouslim, Gérard Le Bihan, and Aurélien Babarit. Le projet SEM-REV : un site d'expérimentation en mer pour la recherche et l'industrie. pages 813–822. Centre Francais du Littoral, 2010. doi: 10.5150/jngcgc.2010.090-r.

[7] Benyamin Schliffke, Sandrine Aubrun, and Boris Conan. Wind Tunnel Study of a "floating" Wind Turbine's Wake in an Atmospheric Boundary Layer with Imposed Characteristic Surge Motion. Journal of Physics: Conference Series, 1618(6), 2020. ISSN 17426596. doi: 10.1088/1742-6596/1618/6/062015.

[8] G. Tarpin. Physical Modelling of Floating Offshore Wind Turbines Inside a Wind Tunnel. Technical report, Centrale Nantes, 2018.

[9] VDI. Environmental meteorology - Turbulence parameters for dispersion models supported by measurement data. Technical report, Verein Deutscher Ingenieure, 2017.

---

## Author Comment (AC2)

**Realistic turbulent inflow conditions for estimating the performances of a floating wind turbine**

C. Raibaudo, J.-C. Gilloteaux, L. Perret

We thank the reviewers for their valuable comments and suggestions. We have addressed all comments and incorporated most suggestions, except when these conflicted between reviewers. Major modifications and additions to the text have been highlighted in blue in the manuscript for better readability.

A rebuttal to the individual reviewer is included below. The number used to identify the reviewer is as provide to us by WES.

**Reviewer #2**

1. A grammatical review would improve the clarity of the paper, with particular emphasis on consistency (RMS and rms are employed variously throughout the manuscript, for example) and employment of past versus present tense.
R1: Sentences have been rephrased, in particular verbs in past.

2. Line 189: The sentence beginning "Compared with typical bluff-body flows, the eigenvalues convergence is slow..." would benefit from the inclusion of numbers to support the comparison.
R2: a sentence and the reference of Leite et al. [2] of the POD performed on the wake of a square cylinder have been added. For this reference, the first two modes corresponds to 53 % of the energy of the velocity fluctuations.

3. The paragraph beginning on Line 214 is interesting and could benefit from additional reasoning and speculation from the authors about the correlations for various modes. The authors provide speculation for the correlations noted for modes 1 and 2 for the spanwise coordinates, but it would be helpful to provide some discussion of the correlations for the wall-normal coordinates.
R3: sentences have been added on the vertical meandering, which could be also the signature of the vertical motion of the wake, and the global decrease of correlation of the amplitude.
Higher modes are usually the combination of multiple motions and therefore difficult to interpret. However, an additional sentence has been added to discuss on these vertical modes (modes 2, 4 and 8) and suggest an interpretation.

4. Line 223: The word "velocities" appears to be missing from the sentence beginning "RMS profiles of the reconstructed along z..."
R4: the typo has been corrected, thanks.

5. Line 230: "loses" instead of "looses"
R5: this typo has been corrected, thank you.

6. Why was 6.6 selected as the tip-speed ratio?
R6: this non-optimal value of TSR has been chosen for consistency with previous (unpublished) work perfomed in our group. It was originally based on the link between the thrust coefficient and the tip-speed-ratio (expressed for example by Eltayesh et al. [1]) .
R6:

7. Figure 10: To improve comparison, it would be helpful to have the same y-axis range for a-2 and b-2.

R7: for all the figures, when the y-range is not the same for subfigures on a same figure, it means their readabilities were not guaranteed, which is the case here.

8. In the Conclusions section, it would be helpful to include a discussion on the authors' speculation on the applicability, potential challenges, and uncertainty of this technique, which is based upon wind tunnel measurements, to a real world operating floating offshore wind farm.

R8: the following paragraph has been added on the perspectives and possible improvements of the technique for floating wind turbines.

*The present study focuses on a limited range of frequencies of motion in the streamwise direction. A possible improvement of the experimental set-up is the consideration of all 6 degrees-of-freedom for the motion and to increase the frequency range for better representativity. For the stochastic tool, classical regularization techniques could help to improve the reconstruction quality. For FAST simulations, it could be interesting to consider floating wind turbines for the model to understand the interaction between the experimental inputs and the FOWT performances simulated by FAST.*

9. Line 333: "allow" instead of "allows"

R9: this has been corrected, thanks.

10. Given that the appendix is quite short, it could easily be incorporated into the main body of the manuscript.

R10: the appendix has been included in the main text.

**References**

[1] Abdelgalil Eltayesh, Magdy Bassily Hanna, Francesco Castellani, A. S. Huzayyin, Hesham M. El-Batsh, Massimiliano Burlando, and Matteo Becchetti. Effect of wind tunnel blockage on the performance of a horizontal axis wind turbine with different blade number. Energies, 12(10):1–15, 2019. ISSN 19961073. doi: 10.3390/en12101988.

[2] Henrique Fanini Leite, Ana Cristina Avelar, Leandra de Abreu, Daniel Schuch, and André Cavalieri. Proper orthogonal decomposition and spectral analysis of a wall-mounted square cylinder wake. Journal of Aerospace Technology and Management, 10:1–15, 2018. ISSN 21759146. doi: 10.5028/jatm.v10.867.